# CPSea: Large-scale cyclic peptide-protein complex dataset for machine learning in cyclic peptide design

**Ziyi Yang[1], Hanyuan Xie[2], Yinjun Jia[3], Xiangzhe Kong[3], Jiqing Zheng[1], Ziting Zhang[4]**

**Yang Liu[3]***, **Lei Liu[1,5,6,7]***, **Yanyan Lan[3,8,9]***

[1]Department of Chemistry, Tsinghua University, Beijing, China
[2]School of Life Sciences, Tsinghua University, Beijing, China
[3]Institute for AI Industry Research (AIR), Tsinghua University, Beijing, China
[4]Department of Automation, Tsinghua University, Beijing, China
[5]Tsinghua-Peking Center for Life Sciences, Beijing, China
[6]Ministry of Education Key Laboratory of Bioorganic Phosphorus Chemistry
and Chemical Biology, Tsinghua University, Beijing, China
[7]Center for Synthetic and Systems Biology, Tsinghua University, Beijing, China
[8]Beijing Frontier Research Center for Biological Structure, Tsinghua University, Beijing, China
[9]Beijing Academy of Artificial Intelligence, Beijing, China

## Abstract

Cyclic peptides exhibit better binding affinity and proteolytic stability compared to their linear counterparts. However, the development of cyclic peptide design models is hindered by the scarcity of data. To address this, we introduce **CPSea**(**C**yclic **P**eptide **Sea**), a dataset of 2.71 million cyclic peptide-receptor complexes, curated through systematic mining of the AlphaFold Database (AFDB). Our pipeline extracts compact domains from AFDB, identifies cyclization sites using the $\beta$-carbon ($C_\beta$) distance thresholds, and applies multi-stage filtering to ensure structure fidelity and binding compatibility. Compared with experimental data of cyclic peptides, CPSea shows similar distributions in metrics on structure fidelity and wet-lab compatibility. To our knowledge, CPSea is the largest cyclic peptide-receptor dataset to date, enabling end-to-end model training for the first time. The dataset also showcases the feasibility of simulating inter-chain interactions using intra-chain interactions, expanding available resources for machine-learning models on protein-protein interactions. The dataset and relevant scripts are accessible on GitHub (https://github.com/YZY010418/CPSea).

## 1   Introduction

Peptides are short chains of amino acids that play diverse roles in biological systems, often acting as signaling messengers or binding agents. Their versatile and tunable properties make them suitable for pharmaceutical applications. The potential of peptide therapeutics has been explored extensively[1, 2]. However, linear peptides often suffer from limitations such as proteolytic instability, short *in vivo* half-life, and poor membrane permeability, which collectively reduce their efficacy.

---

*Correspondence to Yang Liu <liuyang2011@tsinghua.edu.cn>, Lei Liu <lliu@mail.tsinghua.edu.cn>, and Yanyan Lan <lanyanyan@air.tsinghua.edu.cn>

39th Conference on Neural Information Processing Systems (NeurIPS 2025) Track on Datasets and Benchmarks.

Cyclic peptides, peptides with additional covalent links between non-adjacent residues, effectively address these limitations. First, their rigid conformations mitigate excessive flexibility, reducing the entropy penalty upon binding, thereby enhancing affinity. Second, additional constraints confine possible conformations in a smaller state space, increasing binding specificity, and also protecting cyclic peptides against enzymatic degradation. Third, the cyclic topology reduces the hydrodynamic volume and shields polar groups, creating compact structures that traverse cellular membranes more efficiently(Figure 1A). These advantages make cyclic peptides attractive alternatives to linear peptides in drug development[3, 4].

Machine-learning based generative models are emerging as promising approaches for target-conditioned peptide binder design[5, 6]. These models depend on large-scale, high-quality datasets, and the development of models tailored for cyclic peptide is hindered by the limited data. While cyclic peptide databases such as CycPeptMPDB[7] and CyclicPepedia[8] contain thousands of cyclic peptide entries, there are few structure data of cyclic peptide-receptor complexes. To address this, previous studies rely on expedient methods such as post-generation filtering[9] or hard-coded modifications to linear peptide models[10, 6]. Despite their practicality, these approaches face challenges. Post-generation filtering often suffers from low acceptance rate. Hard-coded modifications support limited cyclization types, typically focusing on backbone amide cyclization, while excluding other important cyclization types such as isopeptide and disulfide bonds. Moreover, cyclic peptide structure datasets might still be necessary for these models to generate reasonable cyclic geometries.

Here, we developed an approach to obtain cyclic peptide-protein complex structures from existing protein structure databases by mimicking inter-chain surfaces from intra-protein interactions. By systematic data mining on AlphaFold Database (AFDB), we curated **CPSea** (**C**yclic **P**eptide **Sea**), a dataset of cyclic peptide-protein complexes, consisting of 2.71 million complex structures. These structures exhibit similar properties to experimental data, and possess high structural fidelity, physicochemical plausibility, and synthetic accessibility. We leveraged CPSea to train three cyclic peptide design models from scratch, which generate reasonable structures upon multiple metrics, showcasing the application of the dataset on machine learning.

In summary, main contributions of this work include:

(1) We provided the first large scale cyclic peptide-protein complex dataset, **CPSea**, containing 2.71 million complexes and three linkage types (mainchain, disulfide, and isopeptide).

(2) We proposed a comprehensive set of metrics for peptide-protein complex quality evaluation, focusing on structural fidelity and wet-lab compatibility, which can be employed to evaluate or curate other structure datasets.

(3) We showcased the feasibility of mimicking inter-chain interactions by intra-chain interactions of proteins, and of creating tailored peptide-protein complex structure datasets based on existing single-chain protein databases.

## 2 Related Works

**Peptide design**. Initially, peptide binder design was based on physical approaches, relying on fragment libraries and force fields to explore peptide conformations and binding modes[11]. Recent advancements have shifted toward machine learning frameworks, including autoregressive models (e.g., PepHAR[12]) and flow-matching / diffusion based models (e.g., PepFlow[13], PPFlow[14], DiffPepBuilder[9], PepGLAD[15]). These models have demonstrated success in designing linear peptides but face challenges when applying to cyclic peptides due to data limitation.

**Cyclic peptide design**. Early efforts on cyclic peptide design were also based on fragment libraries and force fields such as anchor extension[16], which restricted peptide geometries to the input scaffold set. Recent works circumvent data scarcity through post-generation filtering (e.g., DiffPepBuilder[9]) or hard-coded constraints (e.g., AfCycDesign[10], RFpeptides[6]). However, such methods exhibit low acceptance rates or limited cyclization types, and models may need fine-tuning with cyclic peptide structures to generate proper conformations.

**Datasets of peptide-protein complex**. Existing peptide-protein interaction datasets primarily focus on linear peptides. Resources including Propedia[17], PepBDB[18], PepBench[15], PepMerge[13], PepBind[19], PepX[20] and PepPC-F[9] provide curated collections of linear peptide and their bind-

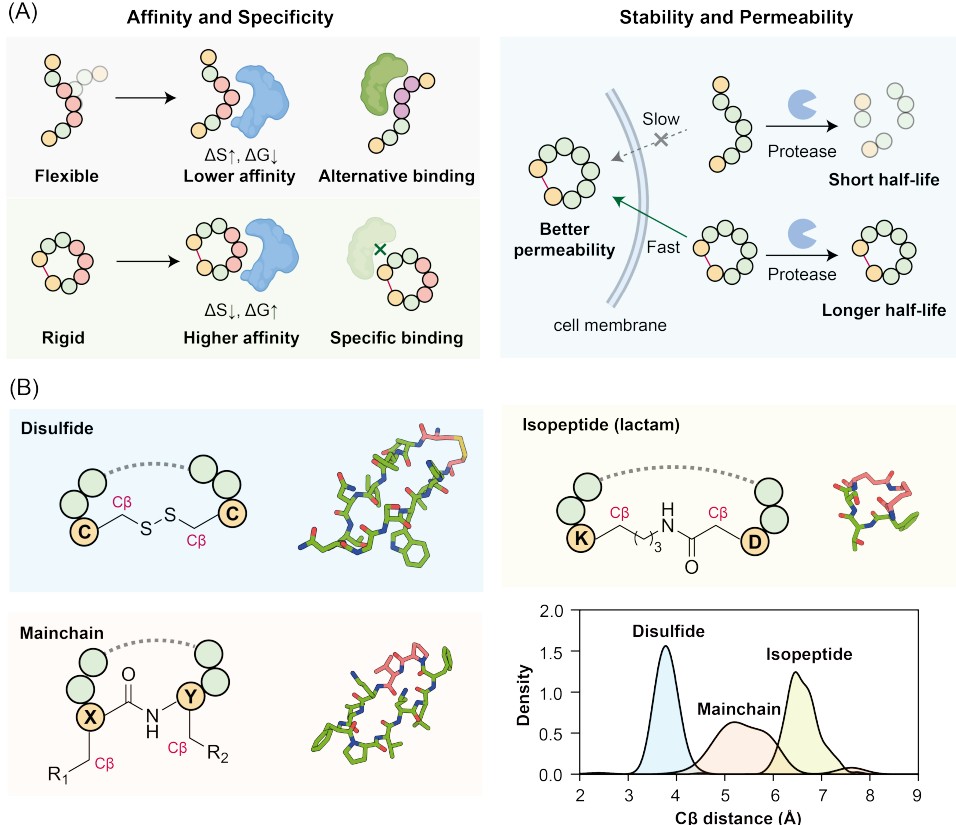

Figure 1: Overall features and $C_\beta$ distance distribution of cyclic peptides. **(A)**: Major advantages of cyclic peptides compared to their linear counterparts. **(B)**: Three common cyclization structures and their $C_\beta$ distance distribution.

ing partners mainly from protein data bank (PDB). For cyclic peptides, CPSet[21] stands out as the biggest dataset, but lacks the data scale and diversity required for model training.

# 3 Our Dataset: CPSea

## 3.1 Data Generation

We started our pipeline by pre-processing AFDB to extract coherent domains with high prediction confidence. We first clustered AFDB entries with a 50% sequence identity threshold, generating a non-redundant database of 8.45 million structures. Structure quality was checked by the predicted Local Distance Difference Test (pLDDT) score[22]. Chains with chain-level mean pLDDT < 69.9 or fewer than 89.9% of residue-level pLDDT $\geq$ 70 were discarded. For each valid chain, we extracted residues with pLDDT $\geq$ 50 and constructed a symmetrized Predicted Aligned Error matrix (PAE matrix) using the minimum of the PAE and its transpose. To avoid sequence-adjacency shortcuts, we masked the near-diagonal band by adding a large penalty. Next, we used agglomerative hierarchical clustering on the PAE matrix with a threshold of 15 Å to identify compact domains. For identified clusters, we split them into contiguous sequence segments and retained segments with more than 10 consecutive residues. Finally, 8.64 million domains were identified (Figure 2).

To detect possible sites for peptide cyclization, we leveraged the distances between $C_\beta$ atoms ($C_\beta$ distances). $C_\beta$ atoms are the first heavy atoms in side chains (except for glycine), providing a geometric anchor for assessing spatial proximity between residues. By surveying experimentally determined protein structures in PDB, we collected structures with disulfide bonds, mainchain amide bonds, and sidechain isopeptide bonds. We observed that these linkages typically constrain corresponding residue pairs to have a $C_\beta$ distance of 3-8 Å (Figure 1B). This empirical distribution was

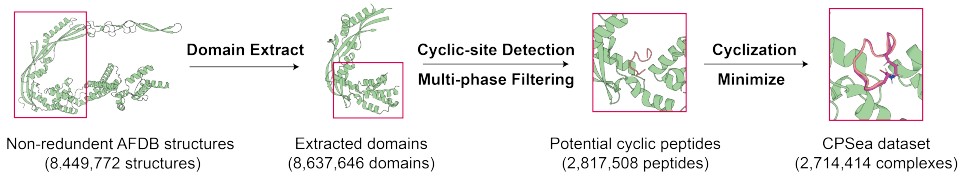



Non-redundant AFDB structures     Extracted domains     Potential cyclic peptides     CPSea dataset
(8,449,772 structures)     (8,637,646 domains)     (2,817,508 peptides)     (2,714,414 complexes)

Figure 2: Data generation pipeline.



used to define potential cyclization sites. It should be noted that this approach is inherently scalable: by tuning the $C_\beta$ distance thresholds, our method can accommodate diverse cyclization chemistries, enabling data generation and model development tailored to other different linkage types.

For each AFDB domain, we generated a $C_\beta$ distance matrix, where each entry represented the $C_\beta$ distance between two residues. Since glycine lacks a side chain and therefore does not have a $C_\beta$ atom, we computed a virtual $C_\beta$ position based on backbone atom positions (N, $C_\alpha$, and C)[23]. Potential cyclic peptides were identified based on the matrix, and a multi-stage filtering protocol was applied to them, as described below (Figure 2):

**Length and breakpoint.** We selected peptides whose lengths were in the range of 5-16. Peptides with non-contiguous residue indices were excluded to ensure backbone continuity.

**Structure quality.** We discarded peptides with a residue-level pLDDT minimum < 70 to ensure structure reliability.

**Disulfide conflict.** To avoid interference of disulfide bonds inside peptides or between peptides and their receptors, we removed candidates with potential non-terminal disulfide bonds. First, we defined receptors as residues whose minimum $C_\beta$ distances to candidate peptides were within 20 Å. Next, we collected all cysteine residues either within peptides (excluding terminal residues for cyclization) or in receptors. If any cysteine pair from these residues had a $C_\beta$ distance < 4.5 Å (typical $C_\beta$ distance for disulfide bonds[9]), and at least one cysteine was in the peptide, the candidate was discarded.

**Secondary Structures.** We focused on $\alpha$-helix and $\beta$-sheet ratios. While these elements are essential for protein folding, cyclic peptides are typically short and structurally constrained, making it less possible to maintain extended $\alpha$-helices or $\beta$-sheets under the unbound state. Moreover, some secondary structures rely on a larger environment to be maintained, such as $\beta$-sheets in $\beta$-barrels, which are hard to preserve without interactions with their neighbors. After manually examining complex structures with different secondary structure percentages, we empirically excluded candidates with >34% $\beta$-sheet or >67% $\alpha$-helix, as these likely represent artifacts of parent protein structures rather than intrinsic features of cyclic peptides themselves.

**Hydrophobicity.** We excluded peptides with >45% hydrophobic residues, because highly hydrophobic peptides are prone to have non-specific hydrophobic interactions in aqueous solutions leading to aggregation or precipitation.

**Binding interface.** We used buried surface area (BSA) to roughly evaluate the binding status between cyclic peptides and their neighbors. BSA is defined as the solvent accessible surface area (SASA) difference between peptides in the bound state and in the unbound state. We applied the following criteria for filtering: (1) Absolute BSA. We retained complex with BSA $\geq$ 400 Å$^2$, ensuring sufficient interfacial contact that reflects binding strength[24]. (2) Relative BSA. We discarded peptides whose relative BSA (rBSA) = BSA/SASA$_{unbound}$ > 85%, to preclude peptides that are buried in the core of a protein, which is unlikely to be the binding mode of cyclic peptides to their receptors. (3) Relative terminal BSA. We strictly constrained terminal residues to have a rBSA < 1%, to avoid disruptions on the binding interface during subsequent terminal mutation and cyclization.

**Receptor connectivity.** We found that some complexes have a "sandwich" structure, where one cyclic peptide interacts with two receptors with separate interfaces. Though these data could be valuable in some applications, we do not want to discuss them in this work. We checked the connectivity of receptor graphs and retained fully connected ones.

Out of 8.64 million AFDB domains, 2.82 million initial cyclic peptides were identified through this filtering process. We then employed OpenMM and PDBFixer to mutate and cyclize terminal residues, and minimized the structure under a modified CharMM36 force field. Since there were

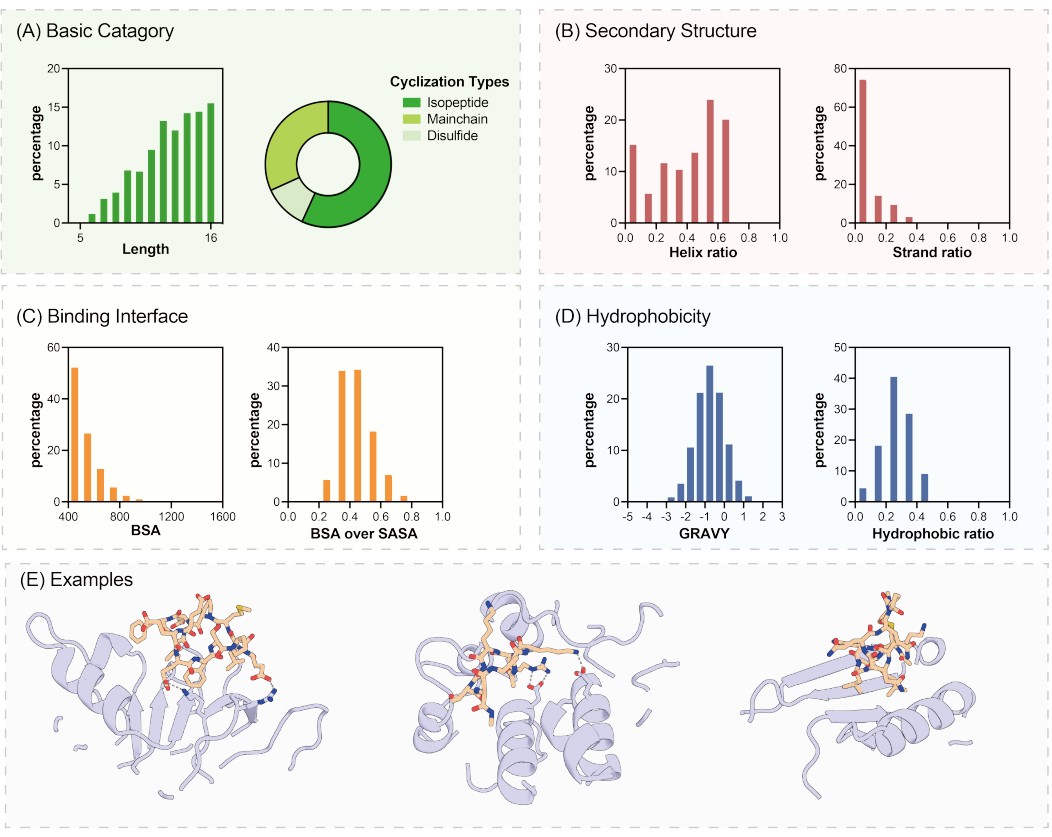

Figure 3: Profiles of CPSea. Distribution on characteristics of CPSea are summarized, and some examples of CPSea complex structures are given.

breakpoints in receptor pockets, we froze these pockets by setting the mass of heavy atoms to zero, and deleted forces corresponding to covalent bonds between heavy atoms. In this way, we restricted heavy atoms to be static, yet allowed hydrogen coordinates to be optimized because the initial hydrogen geometry generated by PDBFixer was always not ideal. After minimization, we applied a filter of minimized energy < 0 to exclude distorted structures, and a chirality filter to ensure all residues in complex structures were of L chirality. Finally, a dataset of 2.71 million cyclic peptide-receptor complex structures was created, which was named **CPSea** (Figure 3).

## 3.2  Data evaluation

### 3.2.1  Metrics

We evaluated the quality of our dataset in three aspects.

**(1) Structure fidelity.** To evaluate whether our synthetic data align with real-world data, we first introduced **Ramachandran plots**, a widely used method in structural biology that encompasses the $\phi$ and $\psi$ dihedral angles of peptide mainchains. Due to steric effect of sidechains, natural proteins adopt certain regions of the $\phi$-$\psi$ plane, known as the favored and allowed regions. Structures are considered plausible if > 95% residues are in the allowed region and > 90% residues are in the favored region[25, 26].

We then considered the proportion of **interaction types** at binding interfaces. We identified interactions by PLIP, which categorizes interactions into types like hydrophobic interactions, hydrogen bonds, salt bridges, etc. The proportion of each interaction type is defined as the number of certain type of interaction over all detected interactions. A similar distribution to native cyclic peptide-protein complexes indicates natural interaction modes.

Finally, we evaluated the **self-consistency** between sequences and structures of cyclic peptides by calculating RMSD between structures in CPSea and structures predicted by HighFold2 based on sequences (which is termed scRMSD). Because it is difficult to predict structures of isopeptide cyclic peptides (Appendix A.2), we only evaluated mainchain and disulfide peptides. We randomly selected 10,000 structures from each type, and reported the average scRMSD. Because receptors in CPSea are discontinuous regions, it is tricky to involve them in the prediction, and only cyclic peptides were refolded in this evaluation.

**(2) Wet-lab compatibility.** We envisioned that one major requirement of peptide design models is that generated peptides should be friendly to downstream wet-lab manipulation (synthesis, biochemical assays, *in vivo* experiments, etc.), which is also a prerequisite for clinical applications. We first calculated **GRAVY** (**GR**and **AV**erage of hydropath**Y**)[27] of each cyclic peptide, a frequently used indicator in chemical protein synthesis. Peptides with positive GRAVY are considered at risk of exhibiting unfavored properties such as low yield in solid-phase synthesis, bad chromatography behaviors, and proneness to aggregation. We then borrowed two indicators in pharmacology: **logP** and **rTPSA**. logP indicates the partition coefficient between octanol and water, related to solubility and membrane permeability. rTPSA is defined as the topological polar surface area (TPSA) divided by the number of heavy atoms, roughly representing the relative abundance of polar structures.

**(3) Diversity and novelty.** Finally, we evaluated the redundancy and novelty of CPSea based on FoldSeek. We clustered CPSea using easycluster-multimer, and reported the diversity as the number of clusters divided by the number of complexes. For novelty, we did FoldSeek multimersearch on CPSea against PDB. For each CPSea complex, we selected the highest qTm value (Tm normalized by query) of cyclic peptides, and calculated their average across all complexes. Since higher Tm indicates higher similarity, we defined novelty as 1-(average highest qTm).

### 3.2.2 Results

To compare *in silico* generated complexes in CPSea with realistic structures, we employed CPSet as a reference for native parameter distributions. The evaluation results are listed below and in Table 1.

**(1) For structure fidelity**, the Ramachandran analysis showed that cyclic peptides in CPSea exhibit reasonable mainchain conformations, with 98.1% of torsions falling within allowed regions, comparable to CPSet's 99.2%. More specifically, 90.1% of torsions occupy favored regions (vs. 94.8% for CPSet), indicating our computational approach effectively preserves natural backbone geometries.

The binding interface analysis revealed slight distribution differences in interaction types. CPSea exhibits 51.8% hydrophobic interactions (vs. 42.0% in CPSet), 38.3% hydrogen bonds (vs. 46.3% in CPSet) and 7.9% salt bridges (vs. 9.0% in CPSet). The shift toward hydrophobic interactions may reflect inherent difference between inter- and intra- protein interactions, but the variance is within an acceptable range, indicating CPSea complexes exhibit plausible binding modes that basically align with experimentally validated structures.

Before evaluating the self-consistency of CPSea, we first benchmarked HighFold2 with 63 experimental cyclic peptide structures from PDB[28], and found an average scRMSD of 2.27 Å. For randomly picked 10,000 mainchain and 10,000 disulfide peptides, the average scRMSD are 2.97 Å and 3.10 Å, relatively higher than those of experimentally determined cyclic peptides, but remain within an acceptable range. One possible reason might be the absence of the receptor context during structure prediction. In summary, these results showed CPSea aligns with experimental data in terms of both structures and interfaces.

**(2) For wet-lab compatibility**, our analysis showed CPSea peptides have an average GRAVY score of -0.74, safely below zero, indicating favorable properties for solid-phase synthesis and low aggregation risk. The average logP value of CPSea peptides (-7.9) was lower than that of CPSet (-5.5), indicating lower hydrophobicity for peptides themselves. The rTPSA metric (5.92) is comparable to experimentally validated structures in CPSet (5.24), further confirmed the appropriately polar nature of CPSea peptides. These results suggest peptides in our dataset are with favorable characteristics for experimental synthesis and potential therapeutic development.

**(3) For diversity and novelty**, FoldSeek generated 617,458 clusters out of 2,714,414 complexes, corresponding to a diversity of 0.227. We looked into the size distribution of clusters, and found that most clusters are of small size, with 77.3% clusters $\leq$ 2, and 94.7% clusters $\leq$ 10. The largest

Table 1: Metrics comparison between CPSea and CPSet

| | Ramachandran | | Interaction Types | | | Hydrophobicity | |
|---|---|---|---|---|---|---|---|
| | Favored | Allowed | Hydrophobic | H-bond | Salt Bridge | logP | rTPSA |
| CPSea | 90.1% | 98.1% | 51.8% | 38.3% | 7.9% | -7.9 | 5.92 |
| CPSet | 94.8% | 99.2% | 42.0% | 46.3% | 9.0% | -5.5 | 5.24 |

cluster consists of 7,925 members, which is only 0.29% of the whole dataset, indicating the cluster distribution is generally balanced. After searching against PDB, the average value of the highest qTm for each structure in CPSea is 0.397, so that the novelty is 0.603, showing that CPSea provides new structures on top of PDB.

## 4 Application on Model development

### 4.1 Creation of specialized subsets

The large scale of CPSea enables us to further divide the dataset into tailored subsets for varied scenarios, where these subsets are still large enough for training specialized models from scratch. We envision three requirements for cyclic peptide design, and created three subsets from CPSea (Figure 4).

**(1) To design binders with high affinity**. Obtaining protein-binding proteins is one of the tasks where *de novo* protein design has shown great potential. Higher affinity is one of the major requirements for protein binders. We employ AutoDock Vina and Rosetta dG for affinity evaluation, and picked complexes that satisfied both Vina score < -6 and Rosetta dG < -25, creating a subset consisting of 545 thousand complexes (20.1% of the CPSea), named **CPBind**.

**(2) To design cyclic peptides with high membrane permeability**. An interesting feature of cyclic peptides is that they exhibit better potential to translocate through membranes, therefore have the potential of oral availability and targeting intra-cellular targets. A representative indicator for membrane permeability is logP, where peptides with logP > -6 are considered possible to show membrane permeability [7]. We applied this simple threshold on CPSea, and identified 799 thousand cyclic peptides.

**(3) To design cyclic peptides with good wet-lab compatibility**. As discussed above, one requirement for cyclic peptide design is that generated candidates should be facile to synthesize and maintain mono-disperse state in solution. We used a GRAVY < 0 filter on CPSea, and found that most peptides (84%) satisfy this filter. We noted that the above selection for membrane permeability prefers peptides that is less hydrophilic, which is antagonistic to wet-lab compatibility. After taking the intersection of the two subsets, a collection of 553 thousand cyclic peptides (20.4% of the CPSea) with both membrane permeability and synthesis compatibility was created, named **CPTrans**.

We then checked whether we can curate a subset that combines all desired features and keeps a scale applicable to model training at the same time. We first took the overlap of CPBind and CP-Trans, generating a subset of 136 thousand complexes. We further checked the physical fidelity by Ramachandran plots, where we seleceted peptides with a Ramachandran accepted ratio higher than 95% and favored ratio higher than 90%. Passing through this filter, a subset of 71,867 complexes is created, which is in a scale comparable to datasets that were used in training linear peptide design models. The subset is named **CPCore**.

### 4.2 Model frameworks

After establishing specialized subsets out of CPSea, we proceeded to validate their utility for training cyclic peptide design models. We roughly regarded cyclic peptides as special linear peptides whose terminal residues are close to each other, and selected three representative frameworks that had been employed in linear peptide design. **DiffPepBuilder** is a SE(3)-equivariant diffusion model trained on PepPC-F, exhibiting good performance in regeneration of known peptide binders and designing novel ligands with improved affinities [9]. **PepFlow** is a model based on conditioned flow matching (CFM) trained on PepMerge, extending CFM to modalities including peptide backbone, sidechain

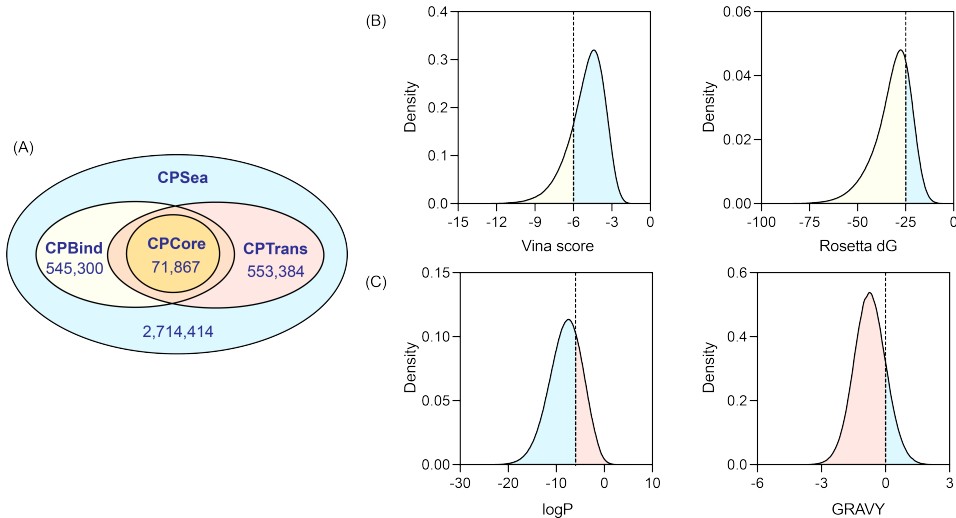

Figure 4: Creation of subsets from CPSea. **(A)**: An overview of the three subsets. **(B)**: Thresholds for curating CPBind. **(C)**: Thresholds for curating CPTrans.

angles and amino acid types. Full-atom peptide design can be accomplished by predicting the joint distribution of flows for these modalities [13]. **PepGLAD** is trained on PepBench and PepBDB, addressing challenges of full-atom geometry and variable binding geometry by using geometric latent space diffusion [15].

## 4.3 Settings

Based on FoldSeek clustering results, there are 30,819 clusters for 71,867 CPCore complexes. We randomly picked 616 clusters (approximately 2%) as valid set and utilized the remaining data as training set. The same data split was applied to all three models.

We employed large non-redundant dataset (LNR) as the test set [29]. After selecting an LNR subset based on ligand length and pocket topology (Appendix B.2), we used MMseqs2 to cluster the remaining 57 target proteins with CPCore, and excluded one target protein that has a sequence identity higher than 40% with CPCore. We generated 100 peptides for each of the remaining 56 targets with the same length as input peptides. The binding epitopes were designated based on original binding interfaces in LNR.

## 4.4 Metrics

**Success rate.** For generated candidates, we checked $C_\beta$ distances of terminal residues, and defined a design "cyclization success" if the distance is between 3-8 Å. For each successful design, we performed the same cyclization and minimization protocol used in our data generation process. We calculated the Rosetta dG for each cyclized complex, based on which we defined "energy success" as Rosetta dG < 0. We also checked the chirality after minimization and defined "chirality success" as peptides consisting entirely of L amino acids. A peptide that meets all success standards was defined as "final success". Only final success peptides were analyzed by the following metrics.

**Structure fidelity.** We employed the same metrics used in dataset evaluation, including Ramachandran plots, interface interaction type distributions and sequence-structure self-consistency based on HighFold2 predictions.

**Wet-lab compatibility and membrane permeability.** We used the same indicators that were employed in subset creation, including GRAVY and logP. According to the cutoffs mentioned above (GRAVY < 0, logP > -6), we reported the ratio of peptides that satisfy the corresponding criteria.

Table 2: Evaluation of models trained on CPSea

| | Success | | | | Affinity | |
|---|---|---|---|---|---|---|
| Models | Cyclization | Chirality | Energy | Final | Rosetta | Vina |
| DiffPepBuilder | 92.2% | 73.9% | **74.8%** | **51.0%** | **-23.3** | -5.9 |
| PepFlow | **93.8%** | 78.2% | 65.0% | 47.7% | -20.8 | **-6.7** |
| PepGLAD | 81.2% | **79.6%** | 70.7% | 45.8% | -23.1 | -6.3 |

| | Interactions | | | Ramachandran | |
|---|---|---|---|---|---|
| Models | Hydrophobic | H-bond | Salt Bridge | Allowed | Favored |
| DiffPepBuilder | 44.1% | 40.8% | 8.3% | **87.0%** | **65.0%** |
| PepFlow | 30.8% | 63.0% | 4.8% | 83.6% | 59.1% |
| PepGLAD | 42.8% | 46.5% | 7.8% | 73.0% | 40.6% |

| | Self-consistency (scRMSD) | | Wet-lab Compatibility | | Diversity and Novelty | |
|---|---|---|---|---|---|---|
| Models | Mainchain | Disulfide | GRAVY | logP | Diversity | Novelty |
| DiffPepBuilder | **2.23** | **2.16** | **83.5%** | **92.9%** | 0.344 | 0.124 |
| PepFlow | 2.26 | 2.55 | 61.4% | 46.5% | 0.578 | **0.138** |
| PepGLAD | 2.53 | 2.50 | 78.9% | 66.0% | **0.598** | 0.120 |

**Binding affinity.** We calculated AutoDock Vina scores and Rosetta dG for each complex. For each target, the peptide with the highest affinity score was selected, and the average values of these best scores were reported.

**Diversity and novelty.** We evaluated the diversity and novelty of the model outputs by the same method used in dataset evaluation based on FoldSeek.

## 4.5 Results

The evaluation results have been summarized in Table 2.

**Success rates.** All three models exhibited a final success rate at about 50%. For cyclization, DiffPep-Builder and PepFlow achieved > 90% success, indicating the potential of generating cyclic peptides. PepGLAD had a lower success rate of 81.2%, because the sidechain idealization process hinders terminal residue sidechains to get close, yet this is essential for proper chirality. For chirality, although DiffPepBuilder and PepFlow are based on SE(3) equivariant networks, and PepGLAD idealizes sidechains after generation, there were still about 20%-25% outputs with wrong chirality. This is mainly because cyclization residues were not specially defined in the vocabulary, and bonds between their sidechains might not form properly during generation. As a result, there were always clashes between terminal residues, leading to high energy states and chirality shifts during minimization. These results indicate that an enlarged vocabulary might be necessary for cyclic peptide design models with better performance.

**Binding affinity.** Although the final success ratios were limited, three models trained on CPSea were still able to design reasonable binding interfaces, with the average Rosetta dG values between -20 and -25 and Vina scores between -5 and -7.

**Structure fidelity.** For Ramachandran plot, PepGLAD struggled to recapture backbone angles, with an allowed ratio of 73.0% and a favored ratio of 40.6%, which is likely because it directly models atom positions rather than sampling torsion angles. For the other two models, the allowed and favored ratios were still lower than the requirements for experimental structures, leaving a space for improving the ability to model backbone dihedral angles. For interaction type distribution, PepGLAD and DiffPepBuilder showed similar distribution to native complexes in CPSet, and PepFlow exhibited a preference for hydrogen bonds. For scRMSD, all three models showed an scRMSD smaller than that of CPSea, with DiffPepBuilder achieved scRMSD at around 2.2 Å. This may be because

models designed cyclic peptides against receptors that have native binders in PDB, and generated peptides were similar to native binders. HighFold2 may be familiar with these interfaces and exhibit a better performance on them. This is consistent with the low novelty for model outputs, as discussed below.

**Wet-lab compatibility** Among three models, DiffPepBuilder showed highest proportion of generated peptides that satisfy GRAVY and logP thresholds, indicating a good ability to learn and sample sequence features.

**Diversity and novelty.** PepGLAD showed the highest structural diversity among three models, which was also higher than the diversity of the training data, indicating the model learned different types of binding structures. For novelty, all three models showed significantly lower novelty compared to that of CPSea. As mentioned before, this is likely because models designed cyclic peptide binders against receptors that have complex structures in PDB, so that the binding interfaces were similar to their native binders. One evidence is that the PDB structure with the highest qTm to a generated cyclic peptide always showed high qTm to the corresponding receptor as well. This is not the case for CPSea, where the interfaces are actually intra-chain interactions from single-chain proteins, so that there are no similar complex structures in PDB.

## 5    Conclusion and limitations

Cyclic peptides are emerging as promising therapeutic candidates due to their advantages in terms of specificity, membrane permeability, etc. However, machine learning approaches to cyclic peptide design have been constrained by data scarcity. To address this challenge, we established CPSea, a dataset of 2.71 million cyclic peptide-receptor complexes derived from the AFDB. A rigorous multistage filtering protocol ensures structure fidelity and biophysical plausibility. The dataset provides the foundation for end-to-end training of cyclic peptide design models.

One major assumption of our work is that intra-chain interactions can be employed to mimic interchain interactions. We note that a similar idea was implemented to construct a domain-domain interaction (DDI) dataset from AFDB as a supplement for PPI datasets, which was used to develop RoseTTAFold2-PPI recently [30]. With the recognition of the feasibility of this approach, the training data for binder design models will be more sufficient through this new data source.

Nevertheless, we acknowledge several limitations in our work:

(1) CPSea captures classical types of cyclization, but has limited coverage of other linkages such as lasso peptides, stapled peptides, and click chemistry derived structures[31, 32]. Nonetheless, as our method is scalable, new types of linkages can be involved in the dataset through similar approaches.

(2) Derived from AlphaFold predictions, CPSea may inherit potential deviation in modeling flexible regions and rare interaction modes. Despite filtering via pLDDT scores and other structural metrics, the extracted binding interfaces may not faithfully capture the physicochemical and geometric features of naturally evolved protein-peptide interaction surfaces.

(3) Due to the large scale of CPSea, we only exhibited model training using a small subset of CPSea. While our experiments demonstrated the utility of our dataset for model training, potential performance improvements from utilizing the whole dataset remains to be further explored.

Despite these limitations, CPSea constitutes a significant advance, offering the largest publicly available dataset of cyclic peptide-protein complexes to date. By releasing this resource to the community, we hope to accelerate progress in AI-driven cyclic peptide binder design.

## Data availability

The datasets and codes for dataset curation, evaluation, model training and evaluation are available at Kaggle, Zenodo and GitHub.

## Acknowledgements

This work is supported by Beijing Academy of Artificial Intelligence and Beijing Frontier Research Center for Biological Structure Fundings, the National Key R&D Program of China (2022YFC3401500), the National Natural Science Foundation of China (T2488301, 22137005, 92253302, 22407121 and 22227810), the National Facility for Translational Medicine(Shanghai) Fundings, the Fundamental Research Funds for the Central Universities, the XPLORER prize, and the New Cornerstone Science Foundation.

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

# A Details in data generation and evaluation

## A.1 Statistical analysis on $C_\beta$ distance distribution

To investigate the $C_\beta$ distance distribution of three cyclization types, we collected structures containing corresponding linkages from Protein Data Bank (PDB) by keyword searching. It should be noted that we used inter-chain isopeptides for estimating $C_\beta$ distance distribution of intra-chain isopeptide bonds, which are theoretically equivalent.

For each structure, we extracted 'SSBOND' lines for disulfides, or 'LINK' lines for mainchain cyclization and isopeptides, and checked the following validation criteria:

(1) Connection residues. For disulfides, the reasonable residue is CYS; for isopeptides, reasonable residues are (LYS, GLU, GLN, ASP, ASN); for mainchain cyclization, reasonable residues are 20 canonical amino acids.

(2) Connection atoms. We checked if the connection atoms are 'C' and 'N' for mainchain cyclization.

Then, we calculated the $C_\beta$ distances between connected residues. For glycines, we added a virtual $C_\beta$ atom based on coordinations of mainchain atoms for the calculation. The same procedure was also applied in other cases where glycines are involved in $C_\beta$ distance calculation:

```
def add_cb(input_array):
    N,CA,C,O = input_array
    b = CA - N
    c = C - CA
    a = np.cross(b,c)
    CB = np.around(-0.58273431*a + 0.56802827*b - 0.54067466*c + CA,3)
    return CB #np.array([N,CA,C,CB,O])
```

A total of 1,823 disulfide bonds, 263 isopeptide bonds, and 173 mainchain cyclization bonds were identified, and the distribution of $C_\beta$ distance was estimated by Kernel Density Estimation (KDE). For isopeptides, we found that most cyclization residue pairs are Lys and Asp/Asn. Therefore, we set connection residues as Lys-Asp/Asn for all cyclic peptides with isopeptide structures during dataset curation and model generation.

## A.2 Filters and metrics

In this section, we describe technical details for filtering and evaluation processes that have not been fully discussed in the main paper, and provide additional examples to illustrate dataset curation in a clearer way.

**Pre-processing on AFDB.** To emphasize the long-distance interactions that define domains, we masked the near-diagonal area of the PAE matrix by adding 35 Å to its values. For the agglomerative hierarchical clustering process, PAE values were used as precomputed distances, and average-linkage clustering with a merge threshold of 15 Å was conducted.

**Receptor definition.** A $C_\beta$ contact matrix was first calculated where element [i, j] is the $C_\beta$ distance between residue i and j. For each residue between potential cyclization sites, we identified matrix elements in the contact matrix with distances less than 20 Å, and excluded residues within 5 residues upstream and downstream of potential cyclization sites. The identified residues were defined as the receptor residues of the cyclic peptide.

**Secondary structure.** The secondary structure was calculated by DSSP based on atom coordinates. Figure 5A shows a frequently observed type of structure discarded in the secondary structure filter, where two strands of a $\beta$-barrel was identified as cyclic peptide, and the rest of the barrel was identified as receptor. However, a $\beta$-barrel with two strands missing is structurally unstable and unlikely to exist independently as an epitope, making the proposed interface implausible.

**Hydrophobic ratio.** Hydrophobic residues were defined as (Val, Ile, Leu, Met, Phe, Trp, Cys). Hydrophobic ratio was calculated as the proportion of hydrophobic residues in cyclic peptides.

**BSA.** The SASA of cyclic peptide in its unbound and bound states were calculated using FreeSASA. Figure 5B shows a representative structure that was filtered out by rBSA threshold, where cyclic

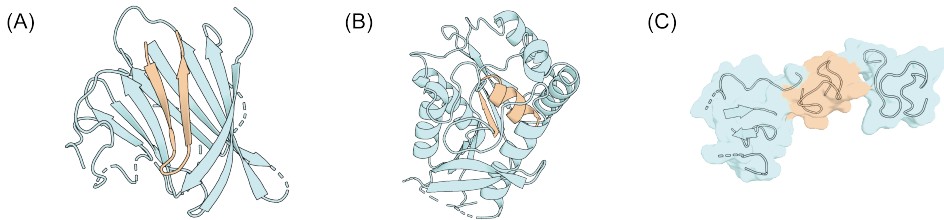

Figure 5: Representative discarded structures. **(A)**: A structure that is filtered out in secondary structure filter. **(B)**: A structure that is filtered out in rBSA filter. **(C)**: A structure that is filtered out in receptor connectivity filter. Cyclic peptides are in wheat color, and receptors are in pale blue.

peptide was surrounded by neighboring residues. Such a binding mode is implausible in nature, as peptides typically cannot penetrate into the interior of receptor proteins.

**Connectivity.** A $C_\beta$ contact matrix of receptor residues was extracted from the global contact matrix, and was binarized based on a connectivity threshold of 10 Å. An undirected graph was constructed where nodes represent residues and edges indicate connectivity, the graph connectivity was checked with NetworkX, and disconnected receptors were discarded. An example of discarded "sandwich" structures is shown in Figure 5C, which is not the scenario we would like to discuss in this work.

**Cyclization and minimize.** Initial cyclic peptides were divided into three linkage types based on $C_\beta$ distances. For terminal residues, their atoms except (N, $C_\alpha$, C, O, $C_\beta$) were removed and they were renamed to residues necessary for the cyclization. In this way, PDBFixer would add missing atoms for the new residue. Afterwards, certain atoms were removed for adding cyclization bonds, and corresponding bonds were added by concatenating CONECT lines in the PDB string. The structure was minimized under CHARMM36 via OpenMM subsequently.

**Ramachandran plot.** The $\psi$ and $\phi$ angles of cyclic peptides were calculated by Biopython. The allowed and favored regions were referred to PyRama. The allowed and favored ratios were calculated for each cyclic peptides as the proportions of $\psi$-$\phi$ angle pairs that are in allowed and favored regions. The average allowed and favored ratios were reported.

**Interaction types.** Interactions between cyclic peptides and receptors were detected by PLIP. The distribution of interaction types was calculated as the number of a certain type of interactions divided by all detected interactions. The average proportions were reported.

**Self-consistency.** Designability is an important metric for structure design models. A structure is defined as designable if there exists a sequence that folds into it. Since our dataset provides both sequence and structure information, and models trained on CPSea are all sequence-structure co-design models, we consider the self-consistency between sequence and structure to be an appropriate metric for evaluating our dataset and the outputs of the models, because structures in self-consistent sequence-structure pairs are inherently designable.

To benchmark the ability of HighFold2, we tested the model with 63 cyclic peptides in PDB, as listed below:

1BH4, 1R1F, 2ATG, 2K7G, 2KUX, 2LWS, 2LYE, 2M78, 2MN1, 2N07, 2NDN, 5KWZ, 6PIN, 6U7R, 7L53, 7M3U, 1DF6, 1VB8, 2B38, 2KCH, 2KVX, 2LWT, 2LYF, 2M79, 2MSO, 2NB5, 2PO8, 5KX1, 6PIO, 6U7S, 7L54, 7RN3, 1HVZ, 1ZA8, 2ERI, 2KNM, 2LAM, 2LWU, 2LZI, 2M9O, 2MT8, 2NDL, 5H1H, 5WOV, 6PIP, 7F32, 7L55, 7S55, 1JBL, 1ZNU, 2GJ0, 2KUK, 2LUR, 2LWV, 2M77, 2MH1, 2MW0, 2NDM, 5H1I, 5WOW, 6U7Q, 7K7X, 7LHC

For three types of cyclic peptide in our dataset, we only evaluated mainchain and disulfide linkages, because refolding cyclic isopeptides turned out to be tricky. We tried AlphaFold3 and Boltz, which accept bond constraints as input features, to refold 100 cyclic isopeptides in CPSea. However, both models fail to form the intended isopeptide bond. For AlphaFold3, the average NZ-CG distance was 12.3 Å (min: 2.8 Å; max: 39.9 Å), far exceeding the typical distance for amide bonds (< 1.4 Å). Interestingly, in 33/100 predictions, the mainchain N-C distance between head and tail showed a distance < 1.4 Å. This proximity might reflect some mode collapse, where formation of amide bonds occurs when two residues get close. Boltz also failed to form isopeptide bonds despite added

Table 3: Training settings.

| Models | Initial Learning Rate | Batch Size |
|---|---|---|
| DiffPepBuilder | 1e-5 | 32 |
| PepFlow | 5e-4 | 12 |
| PepGLAD | 1e-4 | variable |

constraints, with an average NZ-CG distance of 11.0 Å (min: 3.7 Å, max: 29.2 Å). Still, 12/100 predictions similarly showed headtail N-C distance < 1.4 Å. These results indicate that current structure prediction models are still insufficient for cyclic isopeptides.

**Wet-lab compatibility.** GRAVY, logP and TPSA were calculated by RDKit. rTPSA was defined as the ratio of TPSA to the number of heavy atoms.

**Vina score.** We employed the score_only mode of Autodock Vina 1.1.2 to evaluate the affinity without re-docking the complexes. Vinardo was utilized as the score function.

**Rosetta dG.** We employed the Ddg filter in Rosetta Scripts, in which repacking unbound and bound state before energy calculations were disallowed. Following the convention described in the Rosetta documentation, we termed this metric as dG. "ref2015" was used as the score function.

**Diversity and novelty.** We employed FoldSeek for multimer clustering and searching. The command lines were as follows:

```
foldseek easy-multimercluster data output_dir/clu  output_dir --
    multimer-tm-threshold 0.65 --chain-tm-threshold 0.5 --interface-
    lddt-threshold 0.65 --alignment-type 2 --cov-mode 0 --min-seq-id 0
     --threads 32

foldseek easy-multimersearch data pdb output_dir/out output_dir --
    alignment-type 2 --tmscore-threshold 0.0 --max-seqs 1000 --format-
    output query,target,complexqtmscore,complexttmscore,lddt --threads
     48
```

Diversity was simply defined as the number of clusters divided by the number of complexes. For novelty, we calculated the average qTm, where we only involved the ligand chain (L chain), and excluded data when query and target structures were from the same PDB entry.

# B  Details in model training and generation

## B.1  Training.

We kept most parameters the same as their release versions. Some parameters was summarized in Table 3. For PepFlow, a plateau scheduler was applied with a learning rate decay rate of 0.8 and patience of 10. For PepGLAD, a dynamic batch wrapper was applied so that the batch size is variable. All models were trained on a single NVIDIA A800 GPU for approximately 5 days.

## B.2  Generation.

**Test set.** We selected a subset of LNR as test set based on length of the ligand peptide and the binding topology. Some complexes were not suitable for cyclic peptides, because a cyclic topology was suspected to potentially lead to clashes or unstable structures. (Figure 6)

We used MMseqs2 to calculate sequence identity between LNR and CPCore, the command was as follows:

```
mmseqs cluster database output_dir/clusters output_dir --min-seq-id
    0.4 -c 0.95 --cov-mode 1
```

PDB 2BIN in LNR was the only structure that clusters with complexes in CPCore, and was removed from the test set.

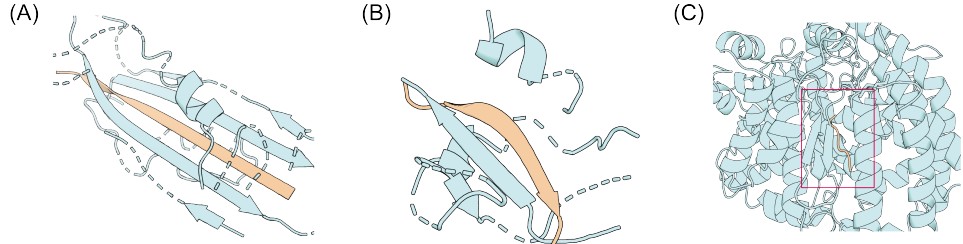

Figure 6: Representative LNR complexes that are not suitable for cyclic peptides. **(A)** and **(B)**: PDB 1JRR and 1SSC. The ligand peptide is a long $\beta$-sheet, which is unlikely to be able to form by cyclic peptides of the same length. **(C)**: PDB 4APH. The ligand peptide is in the compact middle of the protein core. Changing to cyclic topology will likely result in shape mismatch and clash.

Table 4: Data quality of CPSea derived from AFDB and PDB

| | Ramachandran | | Interaction Types | | |
|---|---|---|---|---|---|
| | Favored | Allowed | Hydrophobic | H-bond | Salt Bridge |
| CPSea | 90.1% | 98.1% | 51.8% | 38.3% | 7.9% |
| CPSea_PDB | 87.0% | 97.0% | 48.9% | 39.8% | 9.4% |

| | Wet-lab Compatibility | | | Diversity and Novelty | |
|---|---|---|---|---|---|
| | GRAVY | logP | rTPSA | Diversity | Novelty |
| CPSea | -0.74 | -7.9 | 5.92 | 0.227 | 0.603 |
| CPSea_PDB | -0.95 | -8.2 | 5.93 | 0.475 | 0.639 |

| | Affinity | | Self-consistency(scRMSD) | |
|---|---|---|---|---|
| | Rosetta dG | Vina Score | Mainchain | Disulfide |
| CPSea | $-26.8 \pm 12.3$ | $-5.1 \pm 1.5$ | 2.97 | 3.10 |
| CPSea_PDB | $-31.1 \pm 13.1$ | $-5.1 \pm 1.5$ | 2.98 | 3.07 |

**Pocket designation.** For each complex in LNR dataset, receptor residues within a $C_\beta$-$C_\beta$ distance $< 10$ Å from any residues in peptides were defined as the binding pocket.

**Post generation process.** The process was similar to the curation and evaluation process of CPSea. Briefly, generated peptides whose terminal $C_\beta$ distances were not in between 3-8 Å were excluded, and cyclization types were designated based on $C_\beta$ distances. Terminal residues were mutated, covalent bonds were added. These peptides were then minimized based on a modified Charmm36 forcefield, checked by energy and chirality filters, and evaluated.

## C  Curating dataset from PDB

As discussed in main paper, our pipeline can also be applied to other protein structure databases including PDB. To exemplify this scalability and examine whether using AFDB as data source will introduce biases compared to using experimental structures as data source, we implemented a similar pipeline on PDB.

Briefly, we clustered PDB entries with a 70% sequence identity threshold, and cleaned the structures using Rosetta. The resulting 38,650 entries were processed using a similar cyclic-peptide extraction pipeline as described in the main paper, with the pLDDT filter removed and a amino acid composition filter added (excluding structures with non-canonical modifications or missing atoms). Finally, a dataset of 12,419 cyclic peptide-protein complexes was curated. The dataset is named CPSea_PDB, and is uploaded to Kaggle and Zenodo.

We further explored whether complexes extracted from AFDB and PDB exhibit differences. We employed the same dataset evaluation metrics as described in our manuscript, with results listed in Table 4. Data derived from PDB or AFDB show highly similar results in these analyses, confirming that AFDB is a valid enlarged structural data source that mimics experimental structures for downstream data mining.

