# OpenReview forum: "CPSea: Large-scale cyclic peptide-protein complex dataset for machine learning in cyclic peptide design"
_NeurIPS.cc/2025/Datasets_and_Benchmarks_Track — NeurIPS 2025 Datasets and Benchmarks Track poster_

### Official Review · Reviewer_TKnF · 2025-06-25

**Rating:** 4
**Confidence:** 3

**Summary:**

This manuscript presents CPSea, a large-scale dataset comprising 2.64 million cyclic peptide-protein complex structures systematically curated from the AlphaFold Database (AFDB). The authors describe a computational pipeline for extracting, filtering, and validating cyclic peptide-receptor complexes, and introduce a set of evaluation metrics for assessing structural validity and experimental compatibility. Overall, the work is timely and relevant to the peptide design and computational biology communities.

**Dataset Code Accessibility:**

Yes

**Ethical Considerations:**

No, there are no or only very minor ethics concerns

**Final Justification:**

Considering the rebuttal, I am inclined to maintain my current positive score.

**Limitations Weaknesses:**

1. The dataset is constructed entirely from AlphaFold-predicted structures rather than experimentally determined complexes. Although AlphaFold models are generally reliable, prediction errors, especially for flexible or disordered regions, may propagate into the cyclic peptide dataset.

2. The cyclization detection is primarily based on Cβ–Cβ distances, which may not fully capture more complex or non-canonical cyclization motifs. This could limit the chemical diversity and biological relevance of the generated cyclic peptide structures.

3. The study does not provide validation of the generated dataset using baseline methods. It would be beneficial to curate a small, rigorously selected test set, preferably containing experimentally determined cyclic peptide-receptor complexes, to benchmark the dataset and pipeline.

**Strengths Contributions:**

1. The authors present a well-documented pipeline for extracting, filtering, and validating cyclic peptide-protein complexes from the AlphaFold Database. The workflow is systematic and scalable.

2. CPSea comprises 2.64 million cyclic peptide-receptor complexes.

3. The dataset generation process incorporates multiple layers of quality control—including structure confidence thresholds, backbone continuity, secondary structure filtering, hydrophobicity constraints, and binding interface assessment—ensuring that the resulting complexes are both structurally sound and biophysically plausible.

4. The methodology is described in detail, with clear explanations of each filtering step and the rationale behind parameter choices.

---

> ### Author Rebuttal · Authors · 2025-07-30
>
> # Overall Response
>
> Thank you for your positive comments on our work. As you noted, we have designed numerous filters, both computation- and biochemistry- related, to generate reasonable structures. We understand the concerns about potential biases associated with using AFDB. However, our evaluation results indicate that CPSea structures is similar to real ones, and, to take a step back, our method can be extended to data sources beyond AFDB. Similarly, Cβ is an effective detection metric for the three cyclization types in our work, yet our method can easily be extended to other cyclic peptides types and alternative metrics. Finally, we used CPSet as a baseline for dataset evaluation. We have also added evaluations of the re-trained models using targets from CPSet to complement the LNR test set. We hope this response can address your questions, and if there are any misunderstandings, please let us know.
>
> ---
>
> # Responses to Limitations Weaknesses
> ## Response to W1
>
> **Thank you for your comment**. We recognize that structures from AFDB may have biases. However, we have minimized the introduction of excessive biases during the filtering process, and demonstrated that the generated structures closely resemble real structures by comparing them with experimental data. Furthermore, our pipeline can also generate data from experimental databases, as shown below.
>
> **First**, we applied multi-stage filters to ensure structure quality, wherein we only selected structures will all residue-level pLDDT > 70, to ensure their reliability according to AF2's self-assessment.
>
> **Additionally**, we have compared the structural plausibility and interaction interface patterns between of CPSea structures with those of natural cyclic peptides, demonstrating that CPSea structures are comparable to native ones in terms of their structural characteristics and interaction modes.
>
> **Finally**, as noted in our manuscript, our pipeline can also be applied to other experimental databases such as PDB. To exemplify this scalability and examine whether using AFDB as data source will introduce biases compared to using experimental structures as data source, we implemented a similar pipeline on PDB, as discussed below.
>
> Briefly, we clustered PDB entries at a 70% sequence identity threshold[1], and cleaned the structures using Rosetta. The resulting 38,650 entries were processed using a similar cyclic-peptide extraction pipeline as described in our manuscript, with the pLDDT filter removed and a amino acid composition filter added (excluding structures with non-canonical modifications or missing atoms). Finially, a dataset of 11,511 cyclic peptide protein complexes was curated and uploaded to Kaggle and Zenodo.
>
> We further explored whether complexes extracted from AFDB and PDB exhibit differences. We employed the same dataset evaluation metrics as described in our manuscript, with results listed below. Data derived from PDB or AFDB show highly similar results in these analyses, confirming that AFDB is a valid enlarged structural data source that mimics experimental structures for downstream data mining.
>
> ||Hydrophobic|H-bonds|Salt bridge|Rama Favoured|Rama Allowed|
> |-|:-:|:-:|:-:|:-:|:-:|
> |CPSea_PDB|53.3%|36.1%|8.7%|89.4%|97.8%|
> |CPSea_AFDB|55.7%|34.6%|7.8%|92.6%|98.8%|
>
> ||Rosetta|Vina|GRAVY|logP|rTPSA|
> |-|:-:|:-:|:-:|:-:|:-:|
> |CPSea_PDB|-26.8 ± 12.3|-5.1 ± 1.5|-0.79|-8.2|5.95|
> |CPSea_AFDB|-27.9 ± 9.7|-5.0 ± 1.4|-0.74|-7.9|5.92|
>
> ## Response to W2
>
> **Thank you for your comment**. We acknowledge that the Cβ distance cannot uniquely determine the structure around cyclization sites, but this does not affect its applicability in our paper nor the extensibility of our method.
>
> **First, the Cβ distance serves as a good detection metric for the three cyclization types discussed in our work**. When we looked into the design and synthesis of cyclic peptides, head-tail, disulfide and isopeptides turned out to be the most prevalent cyclization types. Among these, disulfide and isopeptides are connected by side-chain, so the Cβ distance is tightly constrained since Cβ is the starting atom of side-chains. Head-tail cyclic peptides are cyclized by main-chain amides rather than side-chain linkages. However, as shown in Figure 1B, the Cβ distance remains confined to a narrow range. This is likely due to the rigidity of amide bonds and their adjacency to Cβ atoms, limiting adoptable conformations for head and tail residues. In summary, the three most studied cyclic peptides exhibit discrete and narrow Cβ distance distributions, making it a simple yet effective indicator for detecting potential cyclization sites.
>
> **Second, the Cβ distance can be easily replaced with other metrics for alternative cyclization types**. The Cβ distance is an “independent” metric in our pipeline; that is, one can readily substitute other geometric metrics here (for example: CA distance, amide plane angle, etc.) for identifying potential cyclization sites while retaining the rest of the pipeline. Therefore, for cyclization structures whose features are not accurately captured by the Cβ distance, one can first explore the conformation distribution near cyclization sites by QM/MM methods, then select appropriate metrics for cyclization sites detection. After running the pipeline with these new metrics, the filtered peptides can be cyclized with new cyclization structures as long as corresponding MM forcefield files are generated.
>
> n fact, we are currently exploring the use of other geometric metrics to generate data for additional cyclization types, and these efforts will be presented in future work. Thank you again for your insightful comment.
>
> ## Response to W3
>
> **Thank you for your advice**. In fact, we have tried to use the method of baseline to benchmark our dataset and re-trained models in our work.
>
> **First, CPSet is used as a baseline to benchmark our dataset**. CPSet is curated from high-quality crystal structures in PDB, consisting of 493 experimentally determined cyclic peptide protein complex structures[2]. CPSet is regarded as a representative dataset of real cyclic peptide-protein complexes and serves as a benchmark for evaluating the validity of synthesized data in CPSea. As shown in Figure 4, Table 1 and Table 2, the results showed that CPSea structures have similar properties to CPSet in terms of structure validity and interface patterns, demonstrating that CPSea is a feasible enlarged cyclic peptide-protein complex dataset mimicking experimental structures.
>
> **Besides, LNR dataset is employed to benchmark the models re-trained using CPSea**. LNR is a dataset of non-redundant linear peptide protein complexes, curated by domain experts.[3] On top of LNR, to use a complementary test set that is more in line with the application scenarios, we select 18 cyclic peptide protein complexes from CPSet as the test set for the three re-trianed models. Similar to protocols for the LNR test set, we generate 100 cyclic peptides for each target with the same length as the original complexes. Evaluation results on this test set are similar to those on LNR, as listed below:
>
> |Models|Cyclization Success|Energy Success|Final Success|
> |-|:-:|:-:|:-:|
> |DiffPepBuilder|84.8%|71.2%|60.4%|
> |PepFlow|91.0%|61.4%|55.9%|
> |PepGlad|92.0%|75.5%|69.5%|
>
> |Models|scRMSD (headtail)|scRMSD (disulfide)|Div. RMSD|Div. Foldseek|Novelty|
> |-|:-:|:-:|:-:|:-:|:-:|
> |DiffPepBuilder|2.16 Å|2.27 Å|0.631|0.394|0.116|
> |PepFlow|2.20 Å|2.32 Å|0.668|0.577|0.131|
> |PepGlad|2.40 Å|2.63 Å|0.702|0.583|0.112|
>
> |Models|Hydrophobic|H-bonds|Salt bridge|Rama Favoured|Rama Allowed|
> |-|:-:|:-:|:-:|:-:|:-:|
> |DiffPepBuilder|46.4%|39.9%|6.4%|64.0%|85.8%|
> |PepFlow|34.4%|58.5%|4.3%|57.4%|82.5%|
> |PepGlad|42.5%|46.9%|7.3%|31.7%|59.8%|
>
> |Models|Rosetta|Vina|GRAVY|logP|
> |-|:-:|:-:|:-:|:-:|
> DiffPepBuilder|-24.4|-7.1|80.3%|94.9%|
> PepFlow|-21.6|-7.6|81.5%|56.0%|
> PepGlad|-26.3|-6.7|79.6%|70.4%|
>
> ---
>
> # Response to Dataset Code Accessibility
>
> We just would like to notice that we have uploaded our dataset to Zenodo (ID: 16417466) to make the dataset be stored in a more stable/versioned place. We have also uploaded corresponding scripts to GitHub. To make it easy to reproduce our results, we have also included a guide on GitHub detailing how to replicate the generation and evaluation processes described in our paper. We have added cross-links to the other two platforms on Kaggle, Zenodo, and GitHub, enabling easy access to all CPSea-related sites.
>
> ---
>
> # Ref
>
> [1] Gao, B., et al. (2023). ProFSA: Self-supervised Pocket Pretraining via Protein Fragment-Surroundings Alignment. arXiv preprint arXiv:2310.07229.
>
> [2] Zhao, H., et al. (2024). Comprehensive Evaluation of 10 Docking Programs on a Diverse Set of Protein-Cyclic Peptide Complexes. J Chem Inf Model 64(6), 2112–2124.
>
> [3] Tsaban, T., et al. (2022). Harnessing protein folding neural networks for peptide-protein docking. Nat Commun 13(1), 176.

---

> > ### Comment · Reviewer_TKnF · 2025-08-05
> >
> > Thank you for the rebuttal. I am inclined to maintain current score.

---

> > > ### Author Response · Authors · 2025-08-05
> > > **Thank you for your review and constructive advice**
> > >
> > > Dear Reviewer,
> > >
> > > Thank you for taking the time to consider our rebuttal and for your continued feedback on our work. We truly appreciate your insights and constructive advice. Please let us know if there are any further questions.

---

### Official Review · Reviewer_3UjP · 2025-06-27

**Rating:** 4
**Confidence:** 4

**Summary:**

This paper introduces ​​CPSea​​, a dataset of 2.64 million cyclic peptide-protein complexes curated through systematic mining of the AlphaFold Database (AFDB). Data scarcity is one primary limitation for cyclic peptide design. The authors propose a method to extract cyclic peptide-protein complexes from AFDB.

**Dataset Code Accessibility:**

Partly

**Ethical Considerations:**

No, there are no or only very minor ethics concerns

**Final Justification:**

I will increase the score as stated in the comment.

**Limitations Weaknesses:**

1. The analysis of the dataset from the perspective of machine learning is not enough.
2. There is no validation of CPSea’s utility for training ML models. Even preliminary results would help.
3. The reliance on AFDB predictions may introduce biases. The original AF2 model cannot predict cyclic peptide structures. It may not capture some cyclic peptide-protein interactions.
4. In the evaluation, is it possible to use an affinity prediction model or cyclic peptide structure prediction model to validate the extracted complex structures?

**Strengths Contributions:**

1. Leveraging AlphaFold’s predicted structures for extracting cyclic peptide complexes is innovative.
2. The motivation is valid. The scale of the dataset is a bottleneck for ML-driven cyclic peptide design.

---

> ### Author Rebuttal · Authors · 2025-07-30
>
> # Overall Response
>
> Thank you for your comments and your recognition of our motivation and innovation. Regarding concerns about the lack of validation of CPSea’s utility in training ML models, as well as evaluation based on affinity prediction, we have actually completed the relevent analyses and included these in the Appendix. We apologize for not clearly indicated these results in the main text, and have briefly summarized them in this response. Furthermore, we have evaluated CPSea's designability, diversity, and novelty using structure prediction models and Foldseek, aiming to provide a more comprehensive assessment. We have also provided additional explanations below regarding the rationale of using AFDB predictions as the source for data mining. We hope these responses can address your questions. Please let us know if there are any further concerns.
>
> ---
>
> # Responses to Limitations Weaknesses
> ## Response to W1
>
> **Thank you for your comment**. Actually, we have validated the utility of our dataset by using it to train target-conditioned cyclic peptide design models, with these results included in the Appendix.Specifically, we re-trained three peptide design models from scratch using CPSea, and evaluated their performance by metrics including diversity, binding affinity, structural validity and wet-lab compatibility. The results showed that it is feasible to train target-conditioned cyclic peptide design models based on CPSea. Relevant methods and results are briefly summarized below.
>
> **(1) Model frameworks**. We selected three target-conditioned peptide design models: DiffPepBuilder, which uses SE(3)-equivariant diffusion; PepFlow, which relies on conditioned flow matching; and PepGLAD, which employs latent space diffusion.
>
> **(2) Training**. We curated a subset of CPSea, named CPCore, based on metrics related to affinity and membrane permeability, which consists of 51,820 complexes. A random subset of 1,000 complexes was selected as the validation set. Most training configurations were retained as reported, with some different parameters listed below. All models were trained on a single NVIDIA A100 GPU for approximately 4 days.
>
> |Models|Initial learning rate|Batch size|
> |-|:-:|:-:|
> |DiffPepBuilder|1e-5|32|
> |PepFlow|5e-4|12|
> |PepGlad|1e-4|variable|
>
> **(3) Generation**. A subset of LNR[1] was used as the test set. For each target, 100 cyclic peptides were generated with the same length as in original complexes. The binding epitopes were designated based on the original binding interfaces in LNR.
>
> **(4) Evaluation**. We analyzed generated structures in terms of diversity, novelty, self-consistency (designability), structural validity, wet-lab compatibility, binding affinity, as listed below. Taken together, these results indicate that CPSea enables training models from scratch that can generate viable cyclic peptide binders.
>
> |Models|Div. (RMSD)|Div. (Foldseek)|Novelty|scRMSD (Headtail)|scRMSD (Disulfide)|
> |-|:-:|:-:|:-:|:-:|:-:|
> |DiffPepBuilder|0.618|0.409|0.196|2.11 Å|2.18 Å|
> |PepFlow|0.643|0.556|0.193|2.26 Å|2.48 Å|
> |PepGlad|0.680|0.534|0.186|2.43 Å|2.76 Å|
>
> |Models|Hydrophobic|H-bonds|Salt bridge|Rama Favoured|Rama Allowed|
> |-|:-:|:-:|:-:|:-:|:-:|
> |DiffPepBuilder|47.1%|39.1%|7.5%|63.5%|85.6%|
> |PepFlow|34.0%|60.1%|4.1%|56.6%|81.2%|
> |PepGlad|44.1%|45.7%|7.0%|30.9%|57.9%|
>
> |Models|Rosetta|Vina|GRAVY|logP|
> |-|:-:|:-:|:-:|:-:|
> |DiffPepBuilder|-22.2|-6.4|80.8%|92.8%|
> |PepFlow|-20.9|-6.7|79.0%|52.5%|
> |PepGlad|-25.6|-6.4|80.4%|70.2%|
>
> ## Response to W2
>
> **Thank you for your comment**. We recognize that structures from AFDB may have biases and AlphaFold may be less accurate for predicting cyclic peptides due to data scarcity. However, the use of AFDB is justified by the similarity between the interaction interfaces of cyclic/linear peptides and proteins, the potential generalization ability of AlphaFold-series models to cyclic peptides, and relevant screening and evaluation results.
>
> **First**, although cyclic peptides themselves have distinct structural constraints compared to linear peptides, protein-protein interaction interface should have similar patterns. As an example, A screening effort aimed at discovering cyclic peptide binders for the Keap1 ultimately identified cyclic peptides that exhibit a similar interaction pattern to linear peptides.[2] In our pipeline, we minimized the influence of cyclization on interaction interfaces by selecting cyclic peptide with terminal relative BSA < 0.01. Thus, cyclic peptides in CPSea maintain their intra-protein interfaces, which is within the range AF2 can handle. On top of that, a covalent structure is added, so that the interacting conformations could be pre-organized.
>
> **Second**, although AF2 is not trained on cyclic peptide structures, some evidence have shown that its predictive ability generalizes to cyclic peptides to some extent. For example, HighFold, a model that adds cyclic offset on relative position encoding matrix of AlphaFold-Multimer, successfully recapitulates cyclic peptide structure. [3]
>
> **Third**, we acknowledge that AF2 may introduce some biases. This is why we only selected structures with an all residue-level pLDDT > 70, ensuring their reliability based on AF2's self-assessment. Furthermore, we compared the structural plausibility and interaction interface patterns of CPSea structures and those of natural cyclic peptides, demonstrating that CPSea structures are comparable to natural ones in both structural characteristics and interaction modes.
>
> **Finally**, to take a step back, we would like to emphasize the scalability of our pipeline: a similar protocol can also be applied to experimental structure databases such as PDB. We implemented such a pipeline on PDB as follows.
>
> Briefly, we clustered PDB entries at a 70% sequence identity threshold[1], and clean the structures using Rosetta. The resulting 38,650 entries were processed using a similar cyclic-peptide extraction pipeline described in our manuscript, with the pLDDT filter removed and an amino acid composition filter added (excluding non-canonical structures). Finially, a dataset of 11,511 cyclic peptide protein complexes is curated. The dataset has been uploaded to Kaggle and Zenodo.
>
> We further explored whether complexes extracted from AFDB and PDB will show differences. We used the same dataset evaluation metrics as described in our manuscript, with the results listed below. Datasets derived from PDB or AFDB showed similar results in these analyses, confirming that AFDB is a valid enlarged structural data source that mimics experimental structures for data mining.
>
> ||Hydrophobic|H-bonds|Salt bridge|Rama Favoured|Rama Allowed|
> |-|:-:|:-:|:-:|:-:|:-:|
> |CPSea_PDB|53.3%|36.1%|8.7%|89.4%|97.8%|
> |CPSea_AFDB|55.7%|34.6%|7.8%|92.6%|98.8%|
>
> ||Rosetta|Vina|GRAVY|logP|rTPSA|
> |-|:-:|:-:|:-:|:-:|:-:|
> |CPSea_PDB|-26.8 ± 12.3|-5.1 ± 1.5|-0.79|-8.2|5.95|
> |CPSea_AFDB|-27.9 ± 9.7|-5.0 ± 1.4|-0.74|-7.9|5.92|
>
> ## Response to W3
>
> **Thank you for your advice**. In fact, we performed affinity analysis on CPSea and presented the results in the Appendix. For validation using structure prediction models, we calculated scRMSD based on HighFold2. Compared to scRMSD of real cyclic peptides, CPSea structures exhibit similar values, providing new evidence that CPSea can feasibly simulate real data.
>
> **For affinity evaluation**, we used Rosetta and Vina to assess the interface affinity of CPSea complexes. We adopted commonly accepted thresholds (Vina score < -6; Rosetta ddG < -25) to define effective binding, and found that 20.5% and 57.9% of complexes meet the Vina and Rosetta thresholds, respectively. The intersection of complexes satisfying both criteria contains 476,950 structures, which is still large enough to provide value for machine learning scenarios.
>
> **For validation based on structure prediction models**, we used HighFold2 to refold head-tail and disulfide cyclic peptides in CPSea to analyze their sequence-structure self-consistency. We first benchmarked HighFold2 using experimental cyclic peptide structures, yielding an scRMSD of 2.27 Å. We then randomly picked 1000 head-tail and 1000 disulfide peptides from CPSea and CPCore, and calculated their scRMSD. Compared to experimental structures, CPSea and CPCore exhibit relatively higher scRMSD values, but remain within an acceptable range, indicating that CPSea cyclic peptides have substantial self-consistency and thus designability.
>
> ||scRMSD (Head-tail)|scRMSD (Disulfide)|
> |-|:-:|:-:|
> |CPSea|2.99 Å|3.08 Å|
> |CPCore|2.66 Å|2.84 Å|
>
> For cyclic isopeptides, we found that prediction models accepting bond constraint input cannot generate structures with the desired bonds. We attempted to refold cyclic isopeptides from CPSea using AlphaFold3 and Boltz. For AlphaFold3, the average NZ-CG distance is 12.3 Å (min: 2.8 Å; max: 39.9 Å), far exceeding the typical amide bond distance (< 1.4 Å). Boltz generated similar results, where the average NZ-CG distance of 11.0 Å (min: 3.7 Å, max: 29.2 Å).  These results indicate that current structure prediction models are still inadequate for cyclic isopeptides, so we focused on evaluating scRMSD in head-tail and disulfide cyclic peptides.
>
> ---
>
> # Response to Dataset Code Accessibility
>
> We would like to notice that we have uploaded our dataset and scripts to Kaggle, Zenodo, and GitHub. The URL for Kaggle is in our submission, and the ID for Zenodo is 16417466. Links to the other two platforms can be found on any of the sites. CPSea is fully available now.
>
> ---
>
> # Ref
>
> [1] Tsaban, T., et al. (2022). Harnessing protein folding neural networks for peptide-protein docking. Nat Commun 13(1), 176.
>
> [2] Iegre, J., et al. (2023). A cell-active cyclic peptide targeting the Nrf2/Keap1 protein-protein interaction. Chem Sci 14(39), 10800–10805.
>
> [3] Zhang, C., et al. (2024). HighFold: accurately predicting structures of cyclic peptides and complexes with head-to-tail and disulfide bridge constraints. Brief Bioinform 25(3), bbae215.

---

> > ### Comment · Reviewer_3UjP · 2025-08-07
> >
> > I thank the authors for the rebuttal. Although a bit confused, it addressed some of my concerns. My suggestion is that moving the model training and validation from the appendix to the main paper, as it seems that there is still space for it. Another thing about the dataset is that the structure of the data files would be helpful. Currently, there are several large tar files. In anticipation of that, I increase my score.

---

> > > ### Author Response · Authors · 2025-08-07
> > > **We are working on reorganizing CPSea dataset files**
> > >
> > > Dear Reviewer,
> > >
> > > Thank you for your constructive advice.
> > >
> > > We are sorry for the confusion that our original manuscript organization may have caused. We realize that model training and validation are crucial parts of dataset evaluation, and we will rearrange these contents into the main paper in the camera-ready version.
> > >
> > > We understand that better organized file structure will make our dataset more user-friendly. We are working on reorganizing the dataset files hosted on Kaggle and Zenodo, and will finish the work as soon as possible. We hope this will make CPSea easier to use.
> > >
> > > Thank you again for your valuable suggestions. Please share any other concerns you may have at your convenience. We will be glad to hear from you.

---

> > ### Author Response · Authors · 2025-08-08
> > **We have reorganized our dataset on Kaggle**
> >
> > Dear Reviewer,
> >
> > Thank you again for your feedback.
> >
> > To make the structure of our dataset more clear and user-friendly, we have reorganized the content and structure of the dataset.
> >
> > * **We organized the files according to different subsets**. Each subset folder contains an index file, a folder for property files, and structure tar file(s). In this way, users can first check the property files and the index file to quickly get an overview of the dataset, and can conveniently download different subsets.
> > * **We updated the content of property tables**. Property data on different aspects are integrated. Unnecessary information is removed and table headers are updated.
> > * **We updated the dataset description on Kaggle**. A file tree and detailed explanation on the file structure are provided.
> >
> > We hope these updates will enable users to utilize our CPSea in an easier and quicker way.
> >
> > Due to quota limits, Zenodo has not been updated temporarily. We will sync the same content to Zenodo as soon as our application for a quota increase is approved.
> >
> > We invite you to check out our new Kaggle site, and please tell us if there are any further confusion or concerns. Thank you again for the advice.

---

> ### Author Response · Authors · 2025-08-03
> **Clarification on Rebuttal Response Numbering**
>
> Dear Reviewer,​
>
> We would like to sincerely apologize for an error in the numbering of our responses in the rebuttal.
>
> Upon careful review, we noticed that **our first response is actually for addressing both of the first two weakness** (regarding dataset analysis from the ML perspective and validation of CPSea’s utility for ML training), but was mistakenly labeled as a single response to the first weakness.​
>
> This error led to a misalignment in the subsequent numbering: **our second response actually corresponds to the third weakness, and the third response corresponds to the fourth weakness**.​
>
> We deeply regret any confusion this may have caused. To make our response more clear, **we re-organized our response to W1 and W2 as follows**.
>
> **For response to weakness 3**, please refer to "Response to W2" in the original rebuttal.
>
> **For response to weakness 4**, please refer to "Response to W3" in the original rebuttal.
>
> ---
>
> ## Response to W1
>
> ***W1: The analysis of the dataset from the perspective of machine learning is not enough.***
>
> **Thank you for your comment.** We have tried to evaluate the dataset from a machine learning perspective.
>
> **First**, we validated CPSea’s utility for training three ML models, and showed the results in the Appendix, as discussed in the response to W2.
>
> **Second**, based on comments from other reviewers, we have supplemented the dataset evaluation with diversity, novelty, and designability. These additions make the dataset evaluation more aligned with the perspective of machine learning.
>
> ## Response to W2
>
> ***W2: There is no validation of CPSea’s utility for training ML models. Even preliminary results would help.***
>
> **Thank you for your comment**. Actually, we have validated the utility of our dataset by using it to train target-conditioned cyclic peptide design models, with these results included in the Appendix.Specifically, we re-trained three peptide design models from scratch using CPSea, and evaluated their performance by metrics including diversity, binding affinity, structural validity and wet-lab compatibility. The results showed that it is feasible to train target-conditioned cyclic peptide design models based on CPSea. Relevant methods and results are briefly summarized below.
>
> **(1) Model frameworks**. We selected three target-conditioned peptide design models: DiffPepBuilder, which uses SE(3)-equivariant diffusion; PepFlow, which relies on conditioned flow matching; and PepGLAD, which employs latent space diffusion.
>
> **(2) Training**. We curated a subset of CPSea, named CPCore, based on metrics related to affinity and membrane permeability, which consists of 51,820 complexes. A random subset of 1,000 complexes was selected as the validation set. Most training configurations were retained as reported, with some different parameters listed below. All models were trained on a single NVIDIA A100 GPU for approximately 4 days.
>
> |Models|Initial learning rate|Batch size|
> |:-:|:-:|:-:|
> |DiffPepBuilder|1e-5|32|
> |PepFlow|5e-4|12|
> |PepGlad|1e-4|variable|
>
> **(3) Generation**. A subset of LNR[1] was used as the test set. For each target, 100 cyclic peptides were generated with the same length as in original complexes. The binding epitopes were designated based on the original binding interfaces in LNR.
>
> **(4) Evaluation**. We analyzed generated structures in terms of diversity, novelty, self-consistency (designability), structural validity, wet-lab compatibility, binding affinity, as listed below. Taken together, these results indicate that CPSea enables training models from scratch that can generate viable cyclic peptide binders.
>
> |Models|Div. (RMSD)|Div. (Foldseek)|Novelty|scRMSD (Headtail)|scRMSD (Disulfide)|
> |:-:|:-:|:-:|:-:|:-:|:-:|
> |DiffPepBuilder|0.618|0.409|0.196|2.11 Å|2.18 Å|
> |PepFlow|0.643|0.556|0.193|2.26 Å|2.48 Å|
> |PepGlad|0.680|0.534|0.186|2.43 Å|2.76 Å|
>
> |Models|Hydrophobic|H-bonds|Salt bridge|Rama Favoured|Rama Allowed|
> |:-:|:-:|:-:|:-:|:-:|:-:|
> |DiffPepBuilder|47.1%|39.1%|7.5%|63.5%|85.6%|
> |PepFlow|34.0%|60.1%|4.1%|56.6%|81.2%|
> |PepGlad|44.1%|45.7%|7.0%|30.9%|57.9%|
>
> |Models|Rosetta|Vina|GRAVY|logP|
> |:-:|:-:|:-:|:-:|:-:|
> |DiffPepBuilder|-22.2|-6.4|80.8%|92.8%|
> |PepFlow|-20.9|-6.7|79.0%|52.5%|
> |PepGlad|-25.6|-6.4|80.4%|70.2%|
>
> ---
>
> **We apologize again for the error in the numbering**. Please let us know if there are any further questions. Thank you.

---

### Official Review · Reviewer_WK4Z · 2025-06-30

**Rating:** 5
**Confidence:** 4

**Summary:**

Summary: The authors curate a dataset of complexes betweeen proteins and cyclic peptides by identifying relevant interactions and geometries in the AFDB, leveraging patterns such as C-beta distances to identify potential cyclisation sites. They then perform a lot of filtering for quality control, relax the structures and in the end compare how the final dataset looks like in terms of biophysical properties compared to experimental datasets and find that they look similar.

**Additional Feedback:**

[Q1] Peptide protein interactions are generally interesting for design purposes; have the authors considered doing a similar effort without cyclisation, but just identifying linear peptide protein interfaces?

**Dataset Code Accessibility:**

Partly

**Dataset Code Comments:**

The data is fully available on Kaggle, however a more stable/versioned place of storing the data like Zenodo would be helpful.

In addition, the models that were retrained in the paper are not available as baselines in the kaggle repository; if these are to be part of the paper (as I mentioned in my comments above would be helpful), enabling easy reproduction there would be helpful.

**Ethical Comments:**

The authors curate data from publicily available protein datasets, process them and apply retrain existing machine learning models in this new data, none of which raises significant ethical concerns.

**Ethical Considerations:**

No, there are no or only very minor ethics concerns

**Final Justification:**

The authors improved their reasoning for their dataset and added new metrics and analysis showing how their dataset is of good quality and of use to the community; I therefore vote accept.

**Limitations Weaknesses:**

[W1] The authors curate the dataset and train some models on the datasets in the appendix, but do not show comparisons of these models trained on existing datasets such as CPSet. Ideally, these experiments would 1:1 compare models trained on previous datasets and new datasets and then show improvements via the new data. Describing more clearly initial evidence that models trained on that synthetic data actually perform better than models trained on previouos experimental dataset would make the paper a lot stronger.

[W2] The authors curate a large datasets, but size is not everything: one exciting thing about the AFDB was that novel fold classes were identified that were not present in the PDB. Similarly, the authors should investigate whether their large dataset just has “more of the same” types of peptide-protein interfaces (in which case its usefulness for machine learning would be limited) or if there are new types of interfaces in there; potential ways of doing this would be clustering via FoldSeek-Multimer or via PPiRef.

[W3] The authors evaluate their dataset as well as the generations of their trained models based on biophysical properties; however in a real design setting, people often leverage other metrics to filter candidates, such as designability (i.e. scRMSD after refolding with e.g. AF3-like models), diversity and novelty (for definitions of these metrics see for example [1]). The authors should conduct similar analysis on their data to show that the data is useful for training models that can be used in practical design settings with models that allow prediction of cyclic peptides such as AFCyc that they themselves cite.

[1] Geffner, Tomas, et al. "Proteina: Scaling flow-based protein structure generative models." arXiv preprint arXiv:2503.00710 (2025).

**Strengths Contributions:**

Contributions:

[C1]  Curate a larger cyclic peptide dataset

[C2] Create an automatic pipeline that allows future extensions of this dataset

Strengths

[S1] The filtering pipeline is quite extensive, and the comparison to experimental datasets is convincing in its message that the synthetic CPSea dataset looks in distribution similar to experimental datasets.

---

> ### Author Rebuttal · Authors · 2025-07-30
>
> # Overall Response
>
> Thank you for your positive comments, helpful advice and constructive feedback. As you mentioned, we conducted extensive designs for the filter settings, aiming to generate a synthetic dataset that aligns with the distribution of experimental data — one that is not only accepted in the AI community but also recognized by the chemistry & biology communities. We also wish to emphasize the extensibility of our pipeline. In terms of data sources, it can be easily extended to other databases (e.g., PDB), as we have shown below. In terms of cyclization structures, it can also be achieved by modifying the metrics for detecting potential cyclic peptides, as we will demonstrate in future work. Nonetheless, we acknowledge that our dataset evaluation overlooked certain aspects, such as designability, diversity, and novelty. We supplemented these evaluations below. Please point out any additional issues for us, thank you.
>
> ---
>
> # Responses to Limitations Weaknesses
>
> ## Response to W1
>
> **Thank you for your advice.** We acknowledge that a head-to-head comparison between our CPSea and reported datasets would provide more convincing evidence of our data’s advantages and validity. However, there is no existing dataset of cyclic peptide protein complexes that is sufficiently large to train a machine learning (ML) model. To our knowledge, CPSet represents the largest dataset of cyclic peptide protein complexes, yet only 497 complexes are in it. Therefore, we have to evaluate our dataset by analyzing the performance of models trained on CPSea and compare their metrics against conventional standards. In summary, a head-to-head comparison is not feasible because our dataset is the first sufficiently large dataset suitable for ML training.
>
> ## Response to W2
>
> **Thank you for your valuable advice and instructions.** We performed diversity and novelty analyses using Foldseek, and the results indicated that CPSea provides a large number of novel cyclic peptide protein complex structures.
>
> **To assess the diversity of our dataset**, we performed Foldseek-multimer clustering on CPSea and CPCore (multimer Tm threshold = 0.65; chain Tm threshold = 0.5; interface lddt threshold = 0.65). The result showed that there are 574,878 clusters in 2,636,249 CPSea complexes (Div. Foldseek=0.218), and 22,881 clusters in 51,820 CPCore complexes (Div. Foldseek=0.442). We uploaded a list file of representative complexes in each cluster to Kaggle, which can be used as non-redundant subsets of CPSea and CPCore.
>
> **To explore whether our pipeline generates new interface types** not observed in existing cyclic peptide protein complexes, we included CPSet in the clustering process, and set a lenient clustering criterion (multimer Tm threshold=0.1; chain Tm threshold=0.1; interface lddt threshold=0.1), so that all complexes that are similar to CPSet will be put in the same cluster with CPSet members. The result showed there are 219,587 clusters under the criterion, in which 214 clusters have CPSet members, and the remaining 219,373 clusters (2,635,145 members) consist solely of CPSea complexes. This result suggested that most complexes in CPSea exhibit multimeric structural patterns distinct from those observed. This result is somewhat predictable, as the scarcity of experimental data makes it unlikely to cover a wide range of binding interface types.
>
> **For novelty**, we also conducted Foldseek-multimersearch on CPSea against PDB. For each complex, we selected the highest qTm value (Tm normalized by query), and calculated the average across all complexes. The average qTm of CPSea is 0.399, indicating CPSea's novelty relative to PDB, which is consistent with the clustering results above.
>
> ## Response to W3
>
> **Thank you for your advice and kind instructions.** We have recognized the importance of using metrics that are in line with real design scenarios, and thus evaluated our dataset and re-trained models in terms of designability (self-consistency), diversity and novelty.
>
> **(1) For designability**, a structure is defined as designable if there exists a sequence that folds into it. Since our dataset provides both sequence and structure information, and models trained on CPSea are all sequence-structure co-design models, we consider the self-consistency between sequence and structure to be an appropriate metric for evaluating our dataset and the outputs of the models. Specifically, we calculated scRMSD by refolding current sequences instead of generating other sequences by models such as ProteinMPNN. Indeed, if a structure can be folded from its designed sequence, it is designable by definition. In other words, structures in self-consistent sequence-structure pairs are inherently designable.
>
> For head-tail and disulfide cyclic peptides, we used HighFold2, a structure prediction model based on cyclic-offset in position encoding matrix, to refold cyclic peptides from CPSea.[1] We first benchmarked HighFold2 with 63 experimental cyclic peptide structures reported in the HighFold paper, finding an average scRMSD of 2.27 Å, comparatively higher than HighFold’s reported 1.48 Å.[2] However, given that HighFold2 supports disulfide bond conditions, and 2.27 Å is within an acceptable range, we employed HighFold2 for subsequent evaluations.
>
> We randomly picked 1000 head-tail cyclic peptides and 1000 disulfide cyclic peptides from CPSea and CPCore, then refolded them by HighFold2. The average scRMSD values are listed below.
>
> ||scRMSD (Head-tail)|scRMSD (Disulfide)|
> |-|:-:|:-:|
> |CPSea|2.99 Å|3.08 Å|
> |CPCore|2.66 Å|2.84 Å|
>
> The average scRMSD of CPSea and CPCore are relatively higher than those of experimentally determined cyclic peptide structures, but remain within an acceptable range. One possible reason might be the absence of the receptor context during refolding, which is hard to incorporate because the targets in CPSea have breakpoints (ligands and receptors are derived from the same protein).
>
> Refolding cyclic isopeptides turned out to be tricky. We tried AlphdFold3 and Boltz that accept bond constraints as input features on refolding 100 cyclic isopeptides from CPSea. However, both models fail to form the intended isopeptide bond. For AlphaFold3, the average NZ-CG distance was 12.3 Å (min: 2.8 Å; max: 39.9 Å), far exceeding the typical distance for amide bonds (< 1.4 Å). Interestingly, in 33/100 predictions, the main-chain N-C distance between head and tail showed a distance < 1.4 Å. This proximity might reflect some mode collapse, where formation of amide bonds occurs when two residues get close. Boltz also failed to form isopeptide bonds despite added constraints, with an average NZ-CG distance of 11.0 Å (min: 3.7 Å, max: 29.2 Å). Still, 12/100 predictions similarly showed head-tail N-C distance < 1.4 Å. Overall, these results indicate that current structure prediction models are still insufficient for cyclic isopeptides. Therefore, we focused our scRMSD evaluation on head-tail and disulfide cyclic peptides.
>
> We also calculated scRMSD of the cyclic peptides generated by the three re-trained models, the results are listed below:
>
> |Models|scRMSD (Head-tail)|scRMSD (Disulfide)|
> |-|:-:|:-:|
> |DiffPepBuilder|2.11 Å|2.18 Å|
> |PepFlow|2.26 Å|2.48 Å|
> |PepGlad|2.43 Å|2.76 Å|
>
> **(2) For diversity**, we previously clustered outputs of re-trained models by cyclic-aware RMSD. To provide a comprehensive evaluation, we further clustered model outputs by Foldseek-multimer, and calculated the diversity by dividing the number of complexes by the number of clusters. The results are listed below:
>
> |Models|Div. (FoldSeek)|Div. (RMSD)|
> |-|:-:|:-:|
> |DiffPepBuilder|0.409|0.618|
> |PepFlow|0.556|0.643|
> |PepGlad|0.534|0.680|
>
> **(3) For novelty**, we performed a Foldseek-multimersearch of the outputs of re-trained models against the PDB dataset. For each generated structure, we picked the highest qTm value (Tm normalized by query), and calculated the average across all outputs. The results are shown below. We also performed a similar analysis on CPSea, the average highest qTm of CPSea against PDB is 0.399, indicating most structures in CPSea are distinct from those in PDB.
>
> |Models|Novelty|
> |-|:-:|
> |DiffPepBuilder|0.196|
> |PepFlow|0.193|
> |PepGlad|0.186|
> |CPSea|0.601|
>
> ---
>
> # Response to Dataset Code Comments
>
> Thank you for the advice, we have uploaded our CPSea (both from AFDB and PDB) to Zenodo. The Zenodo ID is 16417466.
>
> We have upload re-trained weights to Kaggle, and uploaded corresponding scripts to GitHub. To make it easy to reproduce our results, we have also included a guide on GitHub detailing how to replicate the generation and evaluation processes described in our paper. We have added cross-links to the other two platforms on Kaggle, Zenodo, and GitHub, enabling easy access to all CPSea-related sites.
>
> ---
>
> # Response to Additional Feedback Q1
>
> **Thank you for the advice**. Extracting peptide protein interaction interfaces from AFDB could be possible using a similar pipeline. For example, one could first identify peptides with appropriate length and high pLDDT as candidates, and apply similar filters on peptide properties, binding interfaces, synthesis compatibility, etc. We suppose that this approach could generate a larger database for linear peptide protein complexes compared to existing ones. Nonetheless, since several existing databases of linear peptide-protein complexes already support the development of peptide design models (e.g., PepBench, PepPC-F), curating another new dataset is less appealing for us.
>
> ---
>
> # Ref
>
> [1] Zhu, C., et al. (2025). Predicting the structures of cyclic peptides containing unnatural amino acids by HighFold2. Brief Bioinform 26(3), bbaf202.
>
> [2] Zhang, C., et al. (2024). HighFold: accurately predicting structures of cyclic peptides and complexes with head-to-tail and disulfide bridge constraints. Brief Bioinform 25(3), bbae215.

---

> > ### Comment · Reviewer_WK4Z · 2025-08-05
> >
> > Thanks for addressing my comments, the changes improved the reasoning for theidataset and added new metrics and analysis showing how the dataset is of good quality and of use to the community; I therefore raise my score.

---

> > > ### Author Response · Authors · 2025-08-05
> > > **Thank you for your review and constructive advice**
> > >
> > > Dear Reviewer,
> > >
> > > Thank you for your thoughtful review, detailed feedback, and kind guidance on our work. Your comments and instructions were instrumental in helping us strengthen our work. Please let us know if there are any further questions.

---

### Author Response · Authors · 2025-08-05
**Two days left for discussion, we look forward to your reply**

Dear Reviewers,

Thank you for your constructive feedback.

With only **two days left for the discussion** period (due: August 6, 11:59 PM AoE), we kindly invite you to engage with us in the ongoing discussion.

Your response is crucial for fostering a productive discussion, and it will also help us better understand and examine our work. We would greatly appreciate your insights.

As a brief reminder, we would like to highlight some key points from our rebuttal:​

* We have evaluated the designability, diversity, and novelty of both our CPSea and re-trained models.​
* We applied our pipeline to PDB, providing additional evidence supporting the rationality of using AFDB.​
* We have uploaded the weights of the re-trained models to Kaggle and released our dataset in Zenodo. Besides, scripts and detailed instructions for reproducing our experiments are available on GitHub. Please visit our Kaggle page for links to Zenodo and GitHub.​

For a more detailed discussion on other aspects, please refer to our rebuttal.

Thank you again for your time and valuable insights. **We look forward to your reply**.

---

### Note · Authors · 2025-08-12

Dear Reviewers, AC and SAC,

We would like to express our sincere appreciation for your review and consideration of our work. Building on your insightful comments and generous advice, we have been able to further enhance the quality and value of our work.

**Here, we briefly review some shared concerns:**

***1. The reliability and validity of obtaining cyclic peptide protein complexes using predicted linear protein structures from AFDB.***

* **Theoretically**, cyclic peptides share similar interaction interfaces with linear peptides, and prediction biases could be reduced by selecting high pLDDT regions in AFDB.
* **Experimentally**, CPSea structures show comparable performance to experimental data across multiple evaluations. Besides, cyclic peptides derived from AFDB are similar to those from PDB. These further support the validity of using AFDB as a data source.

***2. The inclusion of additional evaluation metrics.***

* We have supplemented sequence-structure **self-consistency (designability)** assessments based on HighFold2, as well as **diversity** and **novelty** evaluations using Foldseek. These results further validate the plausibility of our structures and the utility of our dataset.

***3. The need for stable storage, clearer structure of the dataset, and easier reproduction of relevant results.***

* We have uploaded CPSea to Zenodo, and reorganized the dataset's file structure to make it more user-friendly.
* We provide scripts on GitHub for dataset evaluation, re-trained models generation, and model outputs evaluation. A detailed protocol for reproduction is also included.

**We have also addressed other concerns and suggestions, including:**

|Concerns/Suggestions|Responses/Supplements|
|:-|:-|
|Compare models trained on previous datasets with the new CPSea|Existing datasets are too small to train ML models|
|Move the model training and validation from the appendix to the main paper|We will reorganize the content in the final version|
|Cyclization detection based on Cβ–Cβ distance may have limitations|Cβ distance is suitable for cyclization structures in this work, and can be easily changed to other metrics for alternative structures|

**At the conclusion of our discussions**, we believe all specifically raised questions and concerns have been addressed, with the reviewers acknowledging our responses. We hope CPSea will facilitate ML model development for cyclic peptide design in the near future.

Thank you for your consideration!

---

### Decision · Program_Chairs · 2025-09-18

**Decision:**

Accept (poster)

**Comment:**

This paper presents the largest cyclic peptide-receptor dataset, enabling training generative models for cyclic peptide design. The authors carefully curated the dataset through systematic mining of the AlphaFold Database and applies multi-stage filtering to ensure structural validity and biding compatibility.  All reviewers agree this is an important contribution to the community and vote for acceptance.